# Solvent–Solute Interaction Effect on Permeation Flux through Forward Osmosis Membranes Investigated by Non-Equilibrium Molecular Dynamics

**DOI:** 10.3390/membranes12121249

**Published:** 2022-12-09

**Authors:** Hayato Higuchi, Masaya Miyagawa, Hiromitsu Takaba

**Affiliations:** Department of Environmental Chemistry and Chemical Engineering, School of Advanced Engineering, Kogakuin University, Tokyo 192-0015, Japan

**Keywords:** forward osmosis, membrane separation, non-equilibrium molecular dynamics, solvation, forward osmosis

## Abstract

The relationship between the solvent–solute interaction and permeation properties is fundamental in the development of the forward osmosis (FO) membrane. In this study, we report on the quantitative reproduction of the permeation flux, which has different solvent–solute interactions, through the modeled FO membrane by non-equilibrium molecular dynamics (NEMD). The interaction effect was investigated by changing the interatomic interaction between the solute and the solvent. The calculated permeation through the semi-permeable modeled FO membrane, in which the interaction between solvent and solution is equal to that between solutions, was consistent with the theoretical curve derived from the combination of the permeation flux and Van’t Hoff equations. These results validate the NEMD for the evaluation of permeation in FO. On the other hand, the permeation is much derived from the theoretical values when the interaction between the solvent and solute atoms is relatively large. However, the simulated permeation was consistent with the theoretical curve, correcting the solution concentration by the coordination number of the solvent atoms to the solute atoms. Our results imply that permeation flux through the FO membrane is significantly changed by the interaction between the solute and the solvent and can be theoretically predicted by calculating the coordination number of the solvent to the solute, which can be readily estimated by equilibrium molecular dynamics simulation.

## 1. Introduction

Much attention has been paid to separation systems that use forward osmosis (FO) membranes because of their high energy efficiency. The water permeation in FO involves the osmotic pressure between two liquid phases of different concentrations without any applied pressure; hence, FO can be applied to various energy-efficient systems, including wastewater treatment [1], and pressure-retarded osmosis (PRO) power plants [2,3]. However, the water permeation flux is considerably slow compared to that of the pressure-induced separation systems using membranes. To obtain the acceptable permeation flux, the choice of solution and fouling resistance of the membranes is fundamentally important since these are the factors that determine the performance of the FO system [4,5]. From this viewpoint, a better understanding of the relationship between the solvent–solute interaction and the permeation property is fundamental.

Several approaches are hitherto proposed to model the permeation properties of FO membranes. One approach is the permeation theory, where the mass transfer of solvent and solute through the FO membrane is analyzed [6]. The experimentally obtained permeation is well reproduced by considering the concentration polarization not only for the membrane, but also its support. Nagy et al. investigated the FO membrane properties used for the PRO [7]. Their models were efficient for macroscopically evaluating the performance of the FO membranes, but the fundamental properties of the membrane on the permeation dynamics must be independently investigated.

Another approach is molecular modeling, which is useful for investigating the correlation between the nanoscopic structure of the membrane materials and their functions. Equilibrium or nonequilibrium molecular dynamics (NEMD) was reported in reverse osmosis and nanofiltration membranes [8,9,10,11,12,13]. In these simulations, a pressure gradient was introduced through a molecular model of the membrane to reproduce the pressure-driven permeation. We developed NEMD using the fluctuating-wall method, which quantitatively reproduces permeation through the nanofiltration membrane [10]. A similar method was applied to various separation systems using a carbon nanotube, graphene, and polymer membranes [11,12,13]. For FO membrane systems, the osmotic pressure is essential for evaluating the permeation flux. A molecular dynamics (MD) study reported that by setting virtual semipermeable membranes in the unit cell, the calculated osmotic pressures obey the values estimated from the Van’t Hoff equation [14]. The MD simulation was further applied to more complex FO systems where the Van’t Hoff equation may not apply. Liu et al. simulated the ion permeation process through a graphene membrane placed in the center of the unit cell. Solutions with different solute concentrations were then placed across the membrane model [15]. In their simulation, a constant pressure was applied to the whole cell to qualitatively investigate the FO membrane dynamics; however, the quantitative analysis of permeation in different solution concentrations was not clearly conducted. Gogoi et al. applied NPT ensemble MD to model the FO using the multi-layer graphene oxide membrane [16]. They successfully investigated the water flux and salt rejection behavior as a function of the distance between the graphene layers. They compared those results with the experimental observation for validation; however, a theoretical validation of the osmotic pressure difference has not been clearly carried out. Indeed, these investigations mainly focus on the qualitative reproducibility of the experimental observation; the effect of the solvent–solute interaction on permeation properties has not been sufficiently discussed yet.

In this paper, we report on the quantitative investigation of permeation flux through the FO membrane by NEMD based on the fluctuating-wall method. Firstly, the permeation results are rationalized by the Van’t Hoff equation, which holds true only for dilute solutions. The Van’t Hoff equation is not applicable in a high-concentration solution or a region where the interaction between solution and solvent is significant. In such conditions, the interaction between the solute and the solvent plays an important role in the nonlinear change in osmotic pressure. By focusing on the solvation structure induced by changing the interaction, we demonstrate that the permeation flux by NEMD can be corrected by the coordination number of the solvent to the solute.

## 2. Calculation Method

The FO membrane process is much slower compared to the RO membrane system because the pressure difference across the membrane is relatively small. Hence, we performed NEMD for a simplified model system of FO. Figure 1a shows a unit cell containing a model of a membrane sandwiched by two fluid phases. The unit cell measured 25.5661 Å × 27.060 Å × 130.000 Å. Periodic boundary conditions were applied, except in the flow direction. Lennard–Jones (LJ)-type fluids were considered in our simulation. Two regions, the pure solvent and solution regions, were prepared by setting a semipermeable membrane between them. Subsequently, 500 LJ fluid atoms were placed in the pure solvent region. This solvent is denoted by ^S^Ar hereafter. In the solution region, 20, 30, 40, or 60 atoms of two different solute atoms, Na and Cl, were placed together with ^S^Ar to model the solutions with 3.5, 4.5, 5.9, or 9.1 wt.%. NaCl was only a model and assumed LJ fluid that dissolved in the solvent as two different solute atoms. The total number of atoms in the solution region was 500. The porous layer was set as the semipermeable membrane between the two regions (Figure 1b). The LJ parameters of the layer atoms were similar to those of the solvent. Its pore diameter was 9.2 Å, which is smaller than the diameter of the solute atoms. The solute and the solution were mixed well and arranged to be stable before starting the permeation in NEMD.

All MDs were performed by NEMD with a fluctuating wall. NEMD with a fluctuating wall reproduced the Hargen–Poiseuille flow through nanoporous membranes [10]. The pressure applied to each solution was independently controlled at 1.0 × 10^−4^ GPa by setting two fluctuating walls placed on both sides of the fluid regions (Figure 1). Two fluid phases were located across a membrane: the solution and the solvent. The solvent and solute atoms provided a 1 force to the fluctuating wall, while a constant external pressure was applied to move the fluctuating walls to balance the force. The velocity-rescaling method was used to maintain the temperature at 0.807 (*T*/*T*_c_), where the unit cell fluid was a liquid [17]. The simulated time was 30 ns with a time step of 1 fs, which was long enough to observe the permeation through the membrane.

Table 1 summarizes the LJ parameters used in this study. A 12-6 LJ potential of $E_{ij}=4\varepsilon_{ij}\left\{{\left(\sigma_{ij}/r_{ij}\right)}^{12}-{\left(\sigma_{ij}/r_{ij}\right)}^{6}\right\}$ was considered, where *i* and *j* denote the atom species, *ε* is the energy parameter, and *σ* is the size parameter of the atom [17]. For the interaction between the heterospecies atoms, $\varepsilon_{ij}$ and $\sigma_{ij}$ were calculated according to the Lorentz–Berthelot combining rules [17]. The atomic charge of all atoms was zero. The *ε* value between the ^S^Ar and solute atoms was doubled to investigate the effect of solvation on FO. The doubled *ε* were chosen so that the flux changes would be distinct. The cutoff distance for the LJ interactions without termination correction was 10 Å.

Permeation was analyzed based on the permeation theory, written as follows:(1)$$J_{w}=L_{p}\left(\sigma_{r}\Delta \pi -\Delta P\right)$$
where *J_w_*, *L*_p_, *σ_r_*, Δ*π*, and Δ*P* represent the flux across the membrane, flux constant, solute reflection coefficient, difference in the osmotic pressure between the solution and solvent phases, and applied pressure difference across the membrane, respectively. Suppose that *σ_r_* is calculated by the Van’t Hoff equation, Equation (1) is written as follows, where Δ*P* is zero for the FO membrane system:(2)$$J_{w}=L_{p}\sigma_{r}\frac{n}{V}RT$$
where *n*, *V*, *R*, and *T* represent the molar concentration of the solute in the solution region, volume of the solution region, gas constant, and temperature, respectively. Considering the changes in the volume of the solution phase in our NEMD, Equation (2) is further modified as follows:(3)$$J_{w}=\frac{nRT}{V+\frac{1}{\rho }\int_{0}^{t}J_{w}^{\prime }S\;dt}L_{p}\sigma_{r}$$
where *ρ*, *S*, and *t* represent the solution density, membrane surface area of the unit cell (25.5661 Å × 27.060 Å), and simulated time, respectively. The *L*_p_ value was determined as 2.91 kg/Pa m^2^ s by fitting Equation (2) to the NEMD results to reproduce the initial flux in NEMD. This coefficient is influenced by the thin membrane thickness. The *σ_r_* value was 1.0, which was adequate because the solute was not able to permeate the membrane. The radial distribution function (RDF) analysis was also conducted to investigate their solvation structure.

## 3. Results and Discussion

### 3.1. Model Validation

Figure 2 shows model snapshots during the NEMD simulation, where the solute concentration was 9.1%. The solute atoms were well distributed over the solution region, and no adsorption on the membrane surfaces was observed. As shown in the figure, apparent external concentration polarization was not observed. Similar distributions were observed for other different concentration or interaction systems. The solvent permeated through the membrane from both phases, but no solute permeated because the pore diameter was smaller than the diameter of the solute. The volume solvent region gradually decreased as time increased. Although the total collision frequency of atoms to the membrane was the same, the difference in the collision frequency of the solvent caused the flux from the solvent phase to the solution phase. This result implies that our NEMD accurately simulated the dynamics of FO membranes.

Figure 3a depicts the relative population changes in the solvent atoms in the solution region during NEMD. *N*_0_ is the initial number of solvent atoms in the solution at 0 fs. *N* is the number of solvent atoms in the solution at each time. The *N*/*N*_0_ values increased as time increased, albeit showing fluctuations. The changes in the low concentrations (i.e., 3.0 and 4.5 wt.%) were similar, and the other two changes in the high concentrations were larger. The *N*_0_ value was different in the concentrations; therefore, the reason behind this trend is unclear. The solvent atom permeation was further analyzed based on the flux.

Figure 3b shows the flux changes during NEMD. The flux was large when the solute concentration was high. Moreover, the simulated data were very consistent with the theoretical curve obtained by Equation (3). This consistency meant that the calculated flux for all concentrations obeyed the Van’t Hoff equation. Δ*π* became large in the high-concentration solution; thus, there was a large amount of flux change at the initial stage of the NEMD calculation. In other words, the gradient was relatively small because Δ*π* was small in the low-concentration solution. In general, the Van’t Hoff equation is not adapted to high concentrations. However, in our calculations, the interaction between the solute atoms and the solvent atoms was the same as that between the solvent atoms. Therefore, the Van’t Hoff equation holds, even in the high-concentration condition. The correlation of calculating flux with the Van’t Hoff equation may deviate when the interatomic interaction between the solute and the solvent becomes large compared to that between the solvents.

### 3.2. Effect of the Interaction between Solute and Solvent on Forward Osmosis

To investigate the effect of the interaction between solute and solvent in the FO, a similar simulation was performed, where the *ε* value of the ^S^Ar solute was doubled. Figure 4a shows the relative population changes in the ^S^Ar atoms in the solution region during NEMD. The *N*/*N*_0_ values increased compared to those shown in Figure 3a. Unlike Figure 3a, the difference in change depending on the concentration was clear. Figure 4a shows that the change in *N*/*N*_0_ becomes small in the range larger than 1.4, which may be due to the contribution of a smaller osmotic pressure difference during the later period of the simulation times. A comparison of Figure 3 and Figure 4 revealed that, more solvent atoms were seemingly permeated from the solvent to the solute phases. The interaction between the solute and the solvent affected the frequency of the collision of the solvent atoms with the membrane surface. Therefore, the microscopic solvation structures related to the intermolecular interaction must be paid further attention.

The RDF analysis clearly revealed the effect of doubled *ε* on the solute environment. Figure 5a,b depict the RDFs of Na with respect to the solvent in the two simulations, where the RDFs of Cl were almost consistent with those of Na. Two peaks appeared around 4.8 and 8.0 Å as the first and second solvation shells, respectively. The first peak intensity was significantly stronger when *ε* was doubled. Thus, more solvent atoms were found to coordinate with the solute atoms. Table 2 summarizes the increased number of coordinating solvent atoms. An increased amount of the coordination number originated from the interaction between the solute and the solvent. Therefore, by doubling the interaction, three or four solvent atoms were affected per solute. These atoms influenced the osmotic pressure. Figure 6 schematically illustrates the difference between the two simulations. More ^S^Ar atoms were weakly bound to the solute due to the strengthened interaction. Thus, fewer solvent atoms were permeated to the solvent region, resulting in a large Δ*π*, followed by an increase in the *N*/*N*_0_ value.

Figure 7 shows the changes in the flux obtained from NEMD, where *ε* is doubled. The fluxes were generally high when *ε* was doubled due to the large Δ*π* and inconsistent with the theoretical curve, as shown in Figure 3b. As described above, this inconsistency originated from the interaction: more solvent atoms were coordinated with the solute. In other words, the number of solvent atoms that behaved as the solvent was decreased. Focusing on this number, the solute concertation correction was carried out by subtracting the number of coordinating solvent atoms in the solute concentration calculation of *n* in Equation (3). When the interaction between the solute and the solvent was ε, the Δπ values for 3.0 wt%, 4.5 wt%, 5.9 wt%, and 9.1 wt% were 1.02 MPa, 1.48 MPa, 1.99 MPa, and 2.66 MPa, respectively. When the interaction was doubled, the corrected Δπ values for 3.0 wt%, 4.5 wt%, 5.9 wt%, and 9.1 wt% were 1.16 MPa, 1.88 MPa, 2.80 MPa, and 5.44 MPa, respectively. Consequently, the flux obtained from NEMD was consistent with the theoretical curves with the concertation correction. Although a certain deviation was observed in the early stage of the NEMD calculation when the concentration was 9.1 wt.%, this was likely because the large difference in osmotic pressure caused a rapid atom movement resulting in flux overshooting. Thus, the solute concentration correction by the coordination number effectively reproduced the data obtained by the NEMD simulation. In other words, NEMD with a fluctuating wall can quantitatively estimate the flux in FO even if the solute concentration effect is significant.

## 4. Conclusions

The fluid permeation in the FO membrane was modeled by a fluctuating-wall NEMD. The calculated fluxes were consistent with the theoretical values delivered from a combination of permeation theory and Van’t Hoff equations, indicating that FO was sufficiently demonstrated. The permeation dependency on the solute concentration was quantitatively rationalized by considering the number of coordinating solvent atoms, which are the function of the strength of the interaction between solute and solvent. The model in this study is rather simplified, but fundamental to the quantitative investigation of the FO membrane systems by NEMD (e.g., complex membranes and multiple solute species). In conclusion, NEMD using a fluctuating wall is effective in further quantitative studies on the permeation flux of more complex structural membranes.

## Figures and Tables

**Figure 1 membranes-12-01249-f001:**
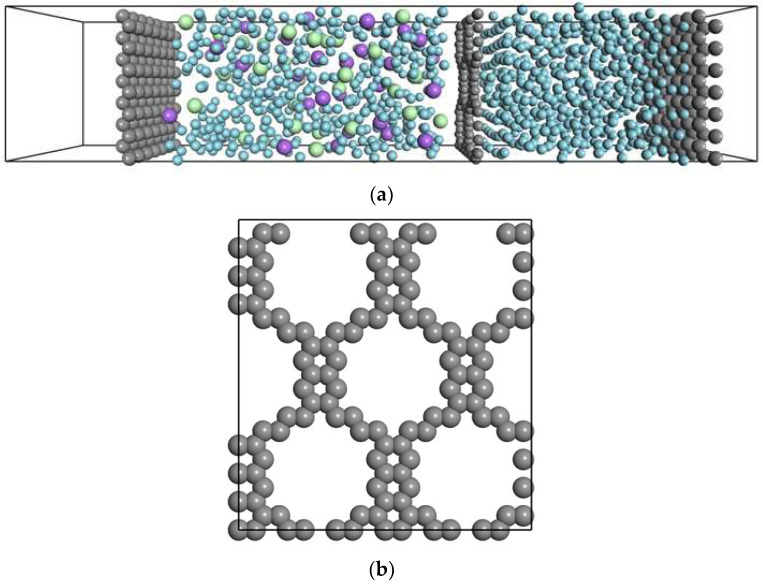
(**a**) Side view of the unit cell for the representative model of FO containing 9.1% solute and (**b**) model of the ^M^Ar membrane. The light blue, gray, green, and purple spheres represent ^S^Ar, ^M^Ar, and Ar wall, the solute (Na), and the solute (Cl), respectively.

**Figure 2 membranes-12-01249-f002:**
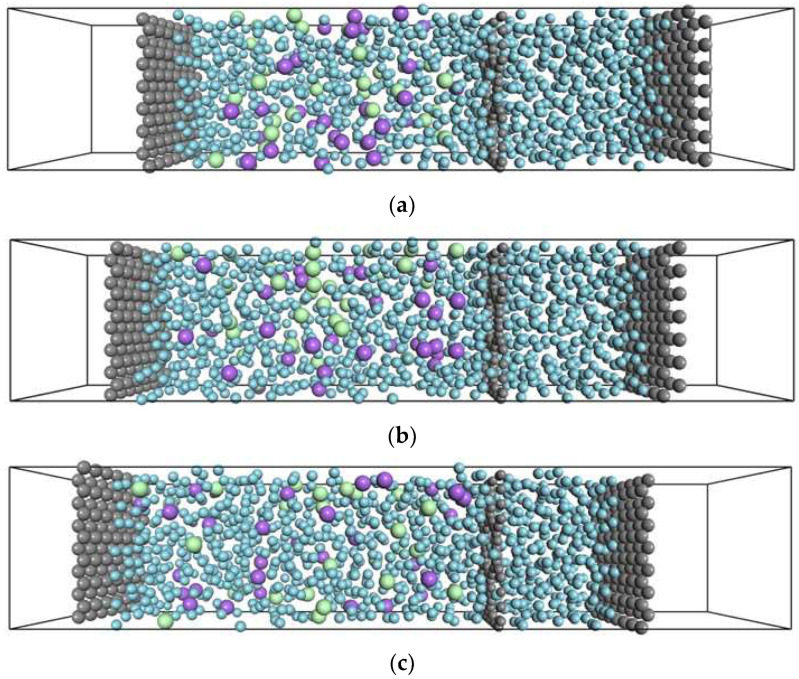
Snapshots of the unit cell from the NEMD results at (**a**) 1, (**b**) 2, and (**c**) 3 ns. The solute concentration is 9.1%.

**Figure 3 membranes-12-01249-f003:**
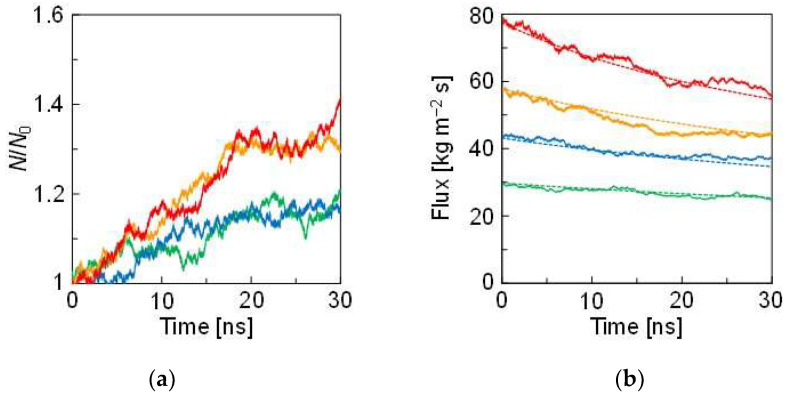
Changes in (**a**) *N*/*N*_0_ and (**b**) the flux during NEMD. The green, blue, orange, and red lines represent 3.0, 4.5, 5.9, and 9.1 wt.% of the solute concentrations, respectively. The solid and dotted lines are obtained by NEMD and Equation (3), respectively.

**Figure 4 membranes-12-01249-f004:**
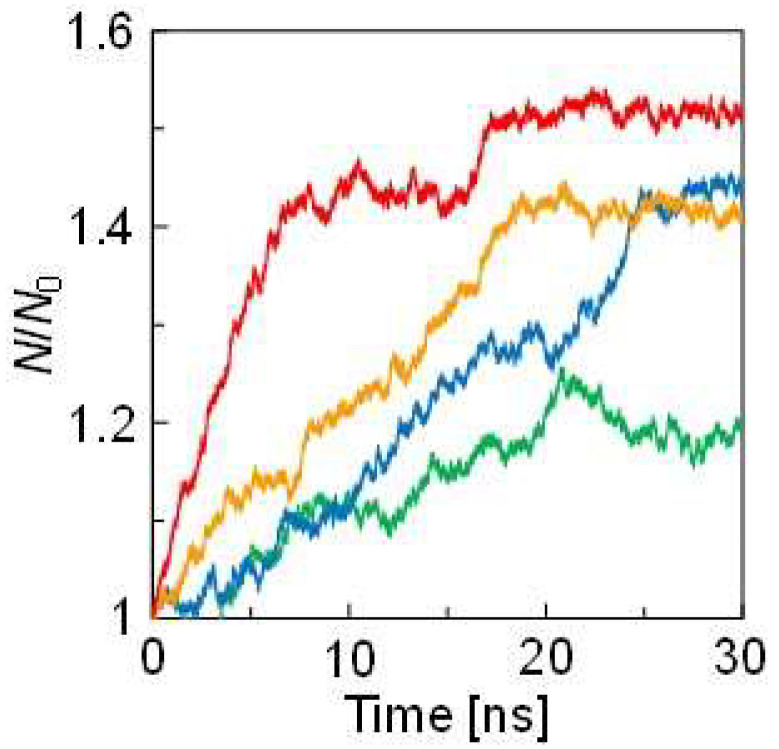
Changes in *N*/*N*_0_ during NEMD, where ε of the ^S^Ar solute is doubled. The green, blue, orange, and red lines represent 3.0, 4.5, 5.9, and 9.1 wt.% of solute concentrations, respectively.

**Figure 5 membranes-12-01249-f005:**
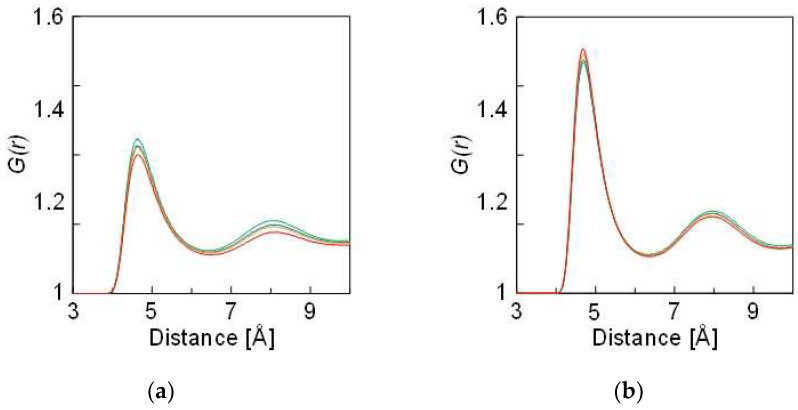
Radial distribution functions of Na with respect to ^S^Ar. The ε value of the ^S^Ar solute is as denoted in Table 1 in (**a**) and doubled in (**b**). The green, blue, orange, and red lines represent 3.0, 4.5, 5.9, and 9.1 wt.% of the solute concentrations, respectively.

**Figure 6 membranes-12-01249-f006:**
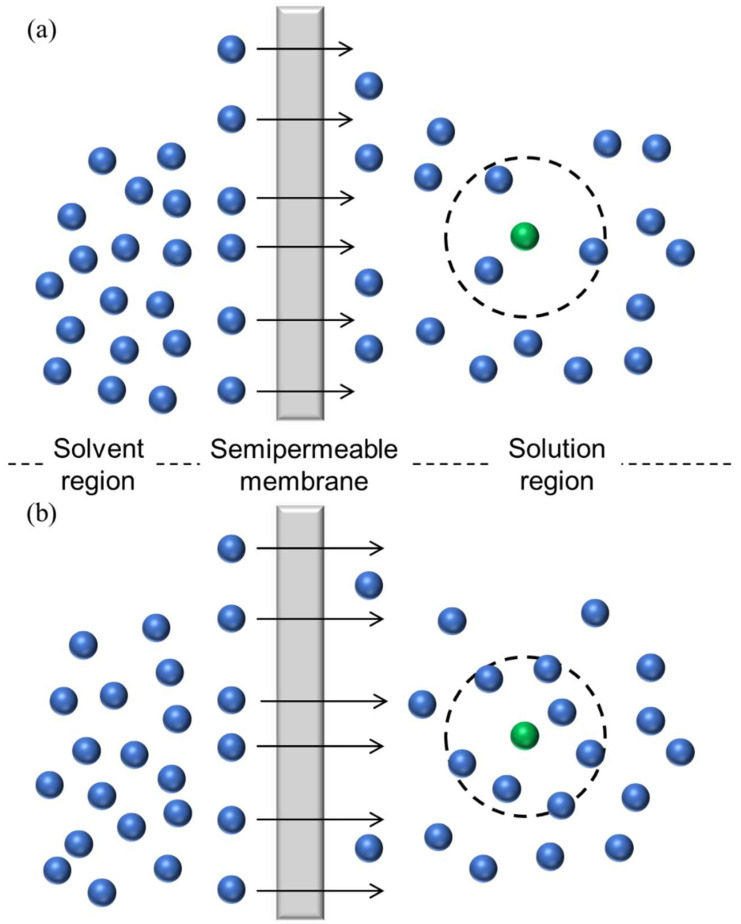
Illustration of the permeation when the ε value of the ^S^Ar solute is as denoted in Table 1 in (**a**) and doubled in (**b**). The blue and green spheres represent the solvent and the solute, respectively.

**Figure 7 membranes-12-01249-f007:**
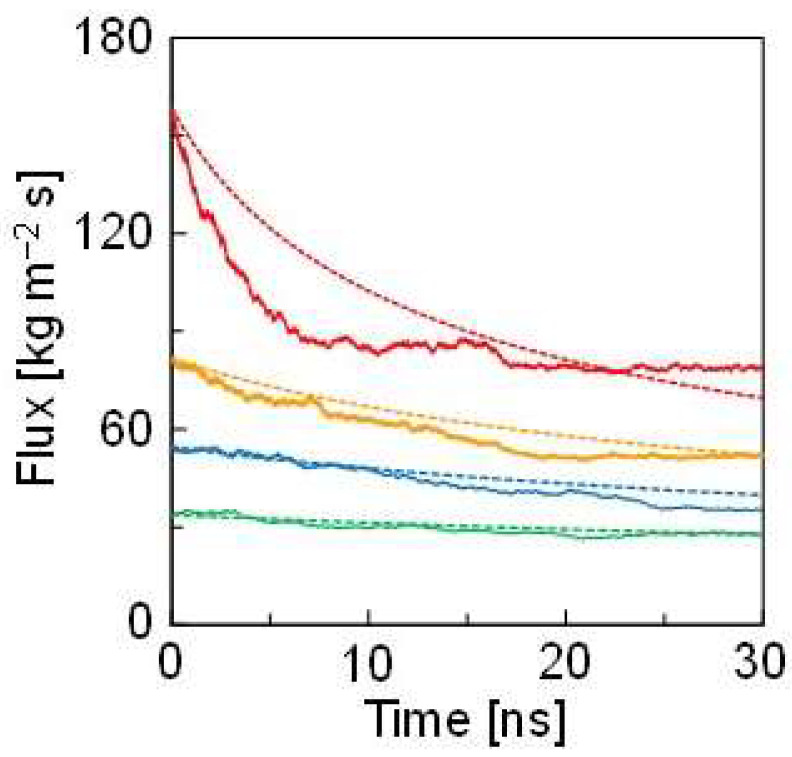
Changes in the flux during NEMD, where the *ε* of ^S^Ar solute is doubled. The green, blue, orange, and red lines represent 3.0, 4.5, 5.9, and 9.1 wt.% of solute concentrations, respectively. The solid lines are obtained by NEMD. The dotted lines are estimated from Equation (3) with the solute concentration correction.

**Table 1 membranes-12-01249-t001:** Lennard–Jones potential parameters used in the present study.

Pair	$\varepsilon_{ii}\;[\times 10^{-21}\;J]$	$\sigma_{ii}\;[Å]$	*Mw* [×10^−3^ kg/mol]
^S^Ar–^S^Ar, ^M^Ar–^M^Ar	1.71	3.42	^S^Ar *=* 39.948
Solute–solute	1.71	5.13	Na = 23.000
			Cl = 35.500

**Table 2 membranes-12-01249-t002:** Calculated coordination numbers of the solvent atom surrounding solute (Na). The values for the *ε* were calculated from the RDFs, as shown in Figure 5a. The values for the doubled *ε* were calculated from the RDFs, as shown in Figure 5b.

Solution	Coordination Number		
	*ε*	Doubled *ε*	Increased Amount
3.0 wt%	13.7	16.7	3.0
4.5 wt%	12.8	16.6	3.8
5.9 wt%	12.3	16.4	4.1
9.1 wt%	11.0	15.4	4.4

## Data Availability

Not applicable.

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
