# Peer review of "Solvent–Solute Interaction Effect on Permeation Flux through Forward Osmosis Membranes Investigated by Non-Equilibrium Molecular Dynamics"

_membranes, 2022, doi:10.3390/membranes12121249_

Round 1

Reviewer 1 Report

The manuscript investigated the effect of solvent-solute interaction on forward osmosis behavior by NEMD simulation. The idea is interesting and the work is concisely presented. It is recommended for publication after several concerns from the reviewer are satisfactorily addressed:

1.     The pore diameter of the membrane is set to be 9.2A, please state the rationale. This pore size is larger than the solute, how does it retain the solute? In line 159, the authors state ‘but no solute permeated because the pore diameter was larger than that of the solute’, please explain the logic.

2.     Please explain how the value of Lp is determined? 2.91kg/Pa m2 s seems to be a very large coefficient. Is it obtained from experimental results?

3.     Do the authors consider external concentration polarization during the simulation?

4.     How large can the change in osmotic pressure be when the interaction is doubled? The percentage? Is it able to account for the change in flux?

5.     In figure 6b, two arrows representing the solvent diffusing from right to left may be removed, since the solvent molecules coordinate with the solute.

6.     The initial flux from the NEMD simulation and mathematical calculation match very well, is it by design? Also, is it possible (i.e. whether extremely long time is needed) to run the NEMD for a longer time interval to check whether the two set of data still match? If not, please at least run the mathematical calculation for longer time (e.g. 1h) based on equation 3 and attach the results in your answer to reviewers’ questions (not necessary to include in the manuscript).

A gentle suggestion: try not to use color to differentiate the curves in the figures; and labeling the varying parameters on the graph can be easier for the readers to follow. 

Author Response

The reply was in the attached file because it includes several figures.

Reviewer 2 Report

The manuscript membranes-2064261 investigated the solvent-solute interaction effect on permeation flux through forward osmosis membranes by non-equilibrium molecular dynamics. The manuscript is helpful for understanding the trans-membrane behavior. The manuscript can be published on Membranes after addressing the following points.

1. The background information of the Lennard–Jones potential parameters are missing. 

2. The ε value of the SAr solute was doubled in order to investigate the interaction between solute and solvent. The reason for the selected parameter should be given.

3. The pore diameter of membrane was larger than the diameter of the solute atoms. However, no solute permeated was observed. The reasons should be discussed.

Author Response

Reply to the referee 2

We appreciate the valuable referee’s comments and questions. We have carefully considered all the comments by the reviewer. We will answer those questions/comments in the order as follows.

Q1.   The background information of the Lennard–Jones potential parameters are missing. 

A1.   We added the equation of a LJ potential in the text in line 11 and the explanation of the parameters in LJ potential was added. Added sentence is “A 12-6 LJ potential of E_ij=4ε_ij {(σ_ij/r_ij )^12-(σ_ij/r_ij )^6 } was considered, where i and j denote the atom species, ε is the energy parameter, and σ is the size parameter of the atom [17]. For the interaction between the heterospecies atoms, ε_ij and σ_ij were calculated according to the Lorentz–Berthelot combining rules [17].”.

Q2.   The ε value of the SAr solute was doubled in order to investigate the interaction between solute and solvent. The reason for the selected parameter should be given.

A2.   The changes were chosen so that the flux changes would be distinct. The reader can easily visualize the effect of a doubled number. We added the reason in the text in line 117: “The changes were chosen so that the flux changes would be distinct.”.  

Q3. The pore diameter of membrane was larger than the diameter of the solute atoms. However, no solute permeated was observed. The reasons should be discussed.

A3.  The sentence in line 159 was miswritten. The size of the solute from the σ, as shown in Table 1, is 5.13*2=10.26A, which was larger than the pore diameter of 9.2A. We corrected the related sentence as follows, “The solvent permeated through the membrane from both phases, but no solute permeated because the pore diameter was smaller than the diameter of the solute.”. We also modified a similar sentence in line 99. We apologize for confusing the referee with this mistake.

Round 2

Reviewer 1 Report

The authors have addressed the comments satisfactorily and the manuscript is recommended for publication.